



# New evidence for atmospheric mercury transformations in the marine boundary layer

Ben Yu[1,2], Lin Yang[1,3], Linlin Wang[1,4], Hongwei Liu[1,3], Cailing Xiao[4], Yong Liang[4], Qian Liu[1], Yongguang Yin[1], Ligang Hu[1], Jianbo Shi[1,2,3*], and Guibin Jiang[1,2,3]

[1]State Key Laboratory of Environmental Chemistry and Ecotoxicology, Research Center for Eco-Environmental Sciences, Chinese Academy of Sciences, Beijing 100085, China.

[2]Institute of Environment and Health, Hangzhou Institute for Advanced Study, University of Chinese Academy of Sciences, Hangzhou, 310000, China.

[3]College of Resources and Environment, University of Chinese Academy of Sciences, Beijing 100049,

China.

[4]Institute of Environment and Health, Jianghan University, Wuhan 430056, China.

*Correspondence to*: Jianbo Shi (jbshi@rcees.ac.cn)

**Abstract.** Marine boundary layer (MBL) is the largest transport place and reaction vessel of atmospheric mercury (Hg). The transformations of atmospheric Hg in MBL are crucial for the global transport and

deposition of Hg. Herein, Hg isotopic signatures in total gaseous mercury (TGM) and particulate bound Hg (PBM) collected during three cruises to Chinese seas in summer and winter were measured to reveal the transformation processes of atmospheric Hg in the MBL. Unlike the observation results at inland sites, isotopic compositions in TGM from MBL were shaped not only by mixing continental emissions, but also largely by the oxidation of $Hg^0$ primarily derived by Br atoms. Lower air temperature could

promote the positive MIF in TGM in summer, while the relative processes might be weak in winter. In contrast, the positive $\Delta^{199}Hg$ and high ratios of $\Delta^{199}Hg/\Delta^{201}Hg$ in PBM indicated that alternative oxidants other than Br or Cl atoms played a major role in the formation of Hg(II) in PBM, likely following the nuclear volume effect. Our results suggested the importance of local Hg environmental behaviours caused by an abundance of highly reactive species, and provided new evidence for understanding the

complicated transformations of atmospheric Hg in the MBL.

## 1 Introduction

The transport and deposition of atmospheric mercury (Hg) are largely attributed to the transformations among three species, including gaseous elemental Hg (GEM), gaseous oxidized Hg (GOM), and particle-bound Hg (PBM), because of the different resident times and migration abilities of



them in atmosphere (Schroeder and Munthe, 1998). Thus, the transformations of atmospheric Hg is

crucial to the global cycling of Hg. The marine boundary layer (MBL) is the largest transport area and

reaction vessel for atmospheric Hg on Earth. It accepts 3400 Mg/yr Hg from ocean via evasion and

deposits 3800 Mg/yr Hg into the ocean (UNEP, 2019). Due to the presence of high relative humidity

(RH), abundant sunshine and atmospheric oxidants, transformations between three species of

atmospheric Hg have been suggested to occur frequently in the MBL (Hedgecock and Pirrone,

2001;Hedgecock and Pirrone, 2004;Laurier et al., 2003;Sprovieri et al., 2010;Wang et al., 2016a;Weiss-

Penzias et al., 2003). Sampling in the MBL provides an opportunity to study atmospheric Hg

transformations, e.g., the scavenging of GEM or the generation of GOM (Hedgecock and Pirrone,

2001;Hedgecock and Pirrone, 2004;Laurier et al., 2003;Sprovieri et al., 2010;Weiss-Penzias et al.,

2003;De Simone et al., 2013;Holmes et al., 2010;Peleg et al., 2015), occurring outside of the influences

of continental emissions. Although oxidizers in the atmosphere, including ozone, hydroxyl radicals,

nitrate radicals, and halogens (Lin and Pehkonen, 1999;De Simone et al., 2013;Holmes et al., 2010;Peleg

et al., 2015;Timonen et al., 2013;Ye et al., 2016;Holmes et al., 2009) have been suggested, the

contributions from multiple redox processes of atmospheric Hg in the MBL have not been clarified. The

mechanisms of the atmospheric Hg transformations in the MBL are also poorly understood.

Compared to other marine studies performed globally, elevated GEM concentrations in the MBL

have been observed in areas of Chinese seas by both coastal and cruise-based observations (Fu et al.,

2010;Wang et al., 2016a;Wang et al., 2016b;Ci et al., 2015;Ci et al., 2011a;Ci et al., 2014;Ci et al., 2011b).

Such observations indicated that anthropogenic emissions from Chinese continental areas impact

atmospheric Hg in the MBL. However, the transformations of atmospheric Hg in MBL are rarely

investigated in these studies.

Stable Hg isotopic method has been utilized to trace the sources and environmental processes of

atmospheric Hg. A ternary system employing mass dependent fractionation (MDF, reported as $\delta^{202}Hg$),

the mass independent fractionation (MIF) of odd isotopes (odd-MIF, reported as $\Delta^{199}Hg$ and $\Delta^{201}Hg$),

and the mass independent fractionation of even isotopes (even-MIF, reported as $\Delta^{200}Hg$), could offer

diagnostic information on the source contributions and historical environmental processes of atmospheric

Hg. In the troposphere, atmospheric Hg isotopes fractionate during the mixing of plumes with various

isotopic compositions, and also during physical processes including volatilization from and dissolution

in droplets (Estrade et al., 2009), diffusion (Koster van Groos et al., 2014), adsorption and desorption on





the airborne particle surfaces (Fu et al., 2019a), and chemical processes (Blum et al., 2014). In addition, odd-MIF suggests occurrence of photo chemical processes when sources mixing can be excluded (Bergquist and Blum, 2007), and even-MIF signatures diagnose the contributions from wet precipitation in natural environment (Enrico et al., 2016).

     Several studies on atmospheric Hg isotopes have conducted at coastal areas, where as the receptors

for mixing air plumes from both continents and the MBL (Demers et al., 2015;Fu et al., 2018;Fu et al., 2019a;Rolison et al., 2013;Yu et al., 2016). These reported isotopic compositions in atmospheric Hg have been suggested as the mixing results of continental anthropogenic emissions and the clean air from MBL. However, the isotopic fractionations occurred during transformations of atmospheric Hg in MBL are diluted by the strong impacts of continental emissions. In order to track the transformations of

atmospheric Hg in MBL using isotopic tracing method, the in-situ sampling is indispensable.

     The objective of this study was to track atmospheric Hg transformations in the MBL using stable Hg isotopes. Both the total gaseous Hg (TGM, composed of GEM and GOM) and PBM samples were collected during three cruises to areas of Chinese seas. The isotopic signatures in TGM and PBM were compared with the observation results at continental sites to extract the fractionations outside of the

influences from anthropogenic emissions, and to reveal the potential mechanisms of the transformation processes of atmospheric Hg in the MBL.

## 2 Materials and Methods

### 2.1 Sample Collection

     The TGM and PBM samples were collected aboard on the Dongfanghong II research vessel during

3 cruises conducted from Jul. 7th to Jul. 20th, 2016 (denoted as 2016-summer cruise), Dec. 29th, 2016 to Jan. 15th, 2017 (denoted as 2016-winter cruise), and Jun. 27th to Jul. 15th, 2018 (denoted as 2018-summer cruise).

     The TGM collection system was constructed following previous publications (Fu et al., 2014;Yu et al., 2016) and using a chloride activated carbon (ClC) trap (Fu et al., 2014) to capture the TGM in ambient

air. A single TGM system was installed on the vessel during the 2016-summer and 2016-winter cruises, and two systems were deployed for the 2018-summer cruise. Two total suspended particle (TSP) collection systems equipped with quartz fiber filter were installed next to the TGM collection systems





for the 2018-summer cruise. Sampling was interrupted during bouts of inclement weather occasionally

experienced during the cruises, and thus was not continuous (Table S1). Sampling durations were divided

into daytime and nighttime periods during the 2016-summer and 2018-summer cruises, and 24h

continuous sampling was conducted during the 2016-winter cruise. See supporting information (SI) for

more details.

### 2.2 Pre-concentration

A thermal-decomposition method using double stage tube furnaces was applied for the pre-

concentrations (Sun et al., 2013;Yu et al., 2016). Acid-trapping solution ($2HNO_3/1HCl$, 40%, v/v) (Sun

et al., 2013) was utilized to capture the released Hg. The Hg concentrations in the trapping solutions were

then measured by cold vapor atomic fluorescence spectrometry following USEPA Method 1631. The

sample solutions with Hg concentrations > 2 ng mL$^{-1}$ were then diluted to 1 ng mL$^{-1}$ to decrease the acid

concentration to < 20% (v/v). Other sample solutions with lower THg concentrations were grouped based

on daytime and nighttime sampling. In each group, the samples were treated with a purge-trap method

using $SnCl_2$ solution and the same acid-trapping solution, and then diluted to 1 ng mL$^{-1}$. The grouped

samples were marked in Table S1. See SI for more details.

### 2.3 Isotopic measurements

Isotopic compositions of the solution samples were measured by a Neptune Plus multi-collector

inductively coupled plasma mass spectrometry using an online vapor generation system (Yin et al., 2010).

The instrument was tuned according to a previous publication (Geng et al., 2018) to obtain high

sensitivity ($^{202}$Hg: 1.6 V per ng mL$^{-1}$ Hg) and steady (internal precision: < 0.1‰) signals. Hg isotopic

compositions were calculated according to the following formulas (Blum and Bergquist, 2007):

$$\delta^{xxx}Hg_{sample} = \left( \frac{^{xxx}Hg_{sample}/^{198}Hg_{sample}}{^{xxx}Hg_{NIST3133}/^{198}Hg_{NIST3133}} - 1 \right) \times 1000 \tag{1}$$

$$\Delta^{xxx}Hg = \delta^{xxx}Hg - \beta \times \delta^{202}Hg \tag{2}$$

$$\beta = \begin{cases} 0.252 & (xxx = 199) \\ 0.502 & (xxx = 200) \\ 0.752 & (xxx = 201) \end{cases} \tag{3}$$

where xxx refers to the mass of each Hg isotope with amu values of 199, 200, 201, and 202.





The 2σ of isotopic compositions for each sample (Table S1) were selected as the greater one of (A) the 2σ of replicated measurements of referenced standards including BCR 482 and SRM 3177, and (B) the 2σ of replicated measurements of each sample.

**2.4 QA/QC**

Data for QA/QC was listed in Table S2.

The performance of the CIC trap was evaluated twice by parallel sampling and by performing breakthrough experiments. Before sampling, ~0.5 g ClC was loaded in each sorbent trap, and the THg content in acid solution was > 10 ng. Thus, the ClC blank counted < 6% (< 2% for most high Hg content samples without merging in pre-concentration step) in all of the acid solution.

BCR 482 (lichen, IRMM, Belgium) was used as the standard to evaluate the recoveries of the pre-concentration procedure. The measured isotopic compositions in the two referenced standards, including BCR 482 and SRM 3177 (mercuric chloride standard solution, NIST) were comparable to reported data (Sun et al., 2016;Estrade et al., 2010). Replicate measurements were conducted during Hg concentration (n = 3) and isotopic measurement (n = 2; except for parallel TGM samples collected in 2018-summer, n = 4). Method blanks were excluded when calculating the Hg concentrations and the pre-concentration recoveries. The mass bias during isotopic measurement was calibrated using sample-standard bracketing method, and using Tl aerosols as an internal spike (Yin et al., 2010).

**2.5 Other supportive data**

The meteorological data collected during the cruise were obtained from an automatic weather station on the Dongfanghong II research vessel.

One of the two parallel sampling filters collected in cruise 2018-summer was treated to measure $Hg^0$, Hg(II), and Br speciation on airborne particles. The measurement of $Hg^0$ and Hg(II) in TSPs were conducted following previous publication (Feng et al., 2004). 1/4 sheet of sampling filter was rolled up and settled in a heating tube installed in tube furnace. The furnace was maintained at 80 °C for 2h and then maintained at 500°C for 2h. Bubbler filled with 5 mL acid-trapping solution same as used in pre-concentration stage were connected at outlet of the heating tube to capture the released Hg at 80 °C and 500°C, respectively. Purified nitrogen used as carrier gas was maintained at 25 mL min-1. The THg in acid-trapping solution was measured using CVAFS following USEPA Method 1631. The other 3/4 sheet





of sampling filter was treated following the National Environmental Protection Standards of the People's Republic of China HJ 799-2016 to obtain the concentration of Br atom, Br anion, and organic Br on TSPs, using ion chromatography.

See SI for more details on the calculation and illustration of 72 h Back-trajectories associated with higher and lower $\Delta^{199}$Hg in TGM using the Hybrid Single-Particle Lagrangian Integrated Trajectory (HYSPLIT4) Model (Rolph et al., 2017;Stein et al., 2015) in Fig. S1.

## 3 Results and discussion

A summary of measured isotopic values and concentrations was listed in Table 1.

### 3.1 Isotopic composition in TGM

According to a William-York bivariate linear regression (Cantrell, 2008) applying $\delta^{202}$Hg and $\Delta^{199}$Hg in all of the TGM samples, the observed fitted curve shaped a slope of -0.10 ± 0.01 (Fig. 1). This fitted curve always indicated a mixing of plumes from anthropogenic emissions characterized by negative $\delta^{202}$Hg and near-zero $\Delta^{199}$Hg values, and plumes from remote areas characterized by positive $\delta^{202}$Hg and negative $\Delta^{199}$Hg values in previous studies (Demers et al., 2015;Yu et al., 2016;Fu et al., 2018;Sun et al., 2014). The eastern region of China is dominated by subtropical monsoon climate, with winds moving from the mainland to the ocean in winter and reversely in summer (Fig. S1). TGM collected during the 2016-winter cruise, that was supposed to be largely impacted by anthropogenic emissions from mainland China based on the monsoon (Fig. S1), but showed positive $\delta^{202}$Hg and negative $\Delta^{199}$Hg ($\delta^{202}$Hg: 0.19 ± 0.30‰; $\Delta^{199}$Hg: -0.13 ± 0.04‰, n = 14, 1σ) echoing the isotopic fingerprints of TGM at the remote sites (Demers et al., 2013;Demers et al., 2015;Fu et al., 2016;Fu et al., 2018;Yu et al., 2016) (Fig. 1). TGM collected during this cruise also showed the highest concentrations (1.81±0.51 ng m$^{-3}$, n = 14, 1σ) among the three cruises, exceeding background value of northern hemisphere (~1.5 ng m$^{-3}$) but falling below averaged GEM concentrations measured at both urban and remote sites in Chinese mainland (urban: 9.20±0.56 ng m$^{-3}$; remote: 2.86±0.95 ng m$^{-3}$) (Fu et al., 2015). Considering that the average wind speed of 6.9 m s$^{-1}$ was measured during this cruise, the air mass leaving Chinese mainland could reach the vessel within several hours. Therefore, isotopic compositions in TGM collected during 2016-winter cruise suggested limited influence from the anthropogenic emissions that diluted in the clean air in MBL.





On the other hand, TGM collected in two summer cruises with larger $\delta^{202}$Hg and $\Delta^{199}$Hg ranges (2016-summer: $\delta^{202}$Hg: -1.48 ± 0.56‰; $\Delta^{199}$Hg: 0.01 ± 0.05‰, n = 9, 1σ; 2018-summer: $\delta^{202}$Hg: -0.09±

0.48‰; $\Delta^{199}$Hg: -0.13 ± 0.06‰, n = 18, 1σ) indicated the mixing of continental emissions to marine originated plumes (Fig. 2 process a). TGM collected in 2016-summer cruise showed near-zero $\Delta^{199}$Hg values, most likely inherited from anthropogenic emissions (Demers et al., 2015;Yu et al., 2016;Fu et al., 2018;Sun et al., 2014), but also showed lower THg concentrations than TGM collected in the other two cruises (Table 1). This result was uncommon because higher TGM concentrations always associated with

anthropogenic emissions in China (Fu et al., 2015). The positive correlation between TGM concentrations and $\Delta^{199}$Hg values in TGM, commonly attribute to mixing of anthropogenic emissions and clean air, was also absent (P > 0.05) in this cruise (Fig. 3b). The back-trajectory analysis results for each cruise also showed the large overlaps of source areas corresponding to plumes with higher and lower $\Delta^{199}$Hg values in TGM (Fig. S1). These uncommon relationships and the large overlaps of source areas

suggested alternative reasons rather than only mixing with continental emissions should contribute to TGM in MBL in summer.

### 3.2 Isotopic composition in PBM

The isotopic compositions in PBM collected from the MBL with negative $\delta^{202}$Hg and positive $\Delta^{199}$Hg values ($\delta^{202}$Hg: -0.80 ± 0.58‰; $\Delta^{199}$Hg: 0.40 ± 0.21‰, n = 9, 1σ) were distinguishable from those

in the TGM (Fig. 1). Similar data have been observed for PBM collected from Huaniao Island, China ($\delta^{202}$Hg: -0.87 ± 0.31‰; $\Delta^{199}$Hg: 0.34 ± 0.34‰) (Fu et al., 2019a). Meanwhile, PBM collected at a coastal site in Grand Bay, USA showed higher $\Delta^{199}$Hg values ($\Delta^{199}$Hg: 0.83± 0.35‰) (Rolison et al., 2013). In contrast, other reported isotopic compositions of PBM almost collected at continental urban/rural sites, have been characterized by negative $\delta^{202}$Hg and near-zero $\Delta^{199}$Hg, due to anthropogenic emissions (Yu

et al., 2016;Das et al., 2016;Huang et al., 2016;Huang et al., 2015;Xu et al., 2017).

The isotopic compositions in PBM in this study and the similar isotopic compositions in PBM collected at island site in China (Fu et al., 2019a) were distinguishable from those collected at inland urban/rural sites, suggesting the dominated influences from marine environment rather than continental anthropogenic emissions. The primary species of PBM examined in this study was Hg(II), accounting

for 78.6±13.0% (1σ, n = 9, Table S3) of total PBM. Therefore, the isotopic fractionations between TGM mostly composed by Hg$^0$, and PBM in MBL, attributed to Hg(II) on the particle surfaces.




Slightly negative $\Delta^{200}$Hg and positive $\Delta^{200}$Hg were observed in both TGM and PBM samples, respectively (Table 1). These near-zero values were comparable to reported data observed in TGM and PBM at most sites globally (Das et al., 2016;Huang et al., 2015;Fu et al., 2019a;Fu et al., 2019b;Fu et al.,

2018;Demers et al., 2015;Demers et al., 2013;Enrico et al., 2016). Higher $\Delta^{200}$Hg values (0.19±0.18‰, 1σ) have been commonly observed globally in wet precipitation with Hg(II) as primary species (Chen et al., 2012;Yuan et al., 2018;Gratz et al., 2010;Enrico et al., 2016;Wang et al., 2015). Those near-zero $\Delta^{200}$Hg values in this study thus indicated the contribution of Hg(II) in wet precipitation to both PBM and TGM should be relatively limited. Therefore, gaseous Hg$^0$ oxidation was implied as an important

contributor to the high $\Delta^{199}$Hg values in PBM.

### 3.3 The oxidation processes in the MBL

MIF induced by the magnetic isotope effect (MIE) mechanism produces a ~1.0 slope in the linear regression of $\Delta^{199}$Hg and $\Delta^{201}$Hg in environmental samples, while a ~1.6 slope is created as a result of the nuclear volume effect (NVE) (Blum and Bergquist, 2007). Although the linear correlation between

$\Delta^{199}$Hg and $\Delta^{201}$Hg in the PBM was insignificant (P > 0.05), ratios of $\Delta^{199}$Hg/$\Delta^{201}$Hg in the PBM samples were significantly higher than 1.0 (Fig. 4), that was the common ratios observed in PBM from sites influenced by anthropogenic emissions (Yu et al., 2016;Das et al., 2016;Huang et al., 2016;Huang et al., 2015;Xu et al., 2017). The observed ratios of $\Delta^{199}$Hg/$\Delta^{201}$Hg in the PBM were also higher than those collected at an island site in China ($\Delta^{199}$Hg/$\Delta^{201}$Hg: ~1.14) (Fu et al., 2019a), and at a coastal site in USA

($\Delta^{199}$Hg/$\Delta^{201}$Hg: ~1.12) (Rolison et al., 2013). The insignificant correlation between $\Delta^{199}$Hg and $\Delta^{201}$Hg in the PBM, and between the isotopic signatures and the percentages of oxidized Hg in the PBM (P > 0.05, Table S3), indicated multiple processes inducing different fractionation rather than single oxidation process occurred.

To date, few isotopic studies have been performed on isotopic fractionation during GEM oxidation,

and the mechanism has been suggested to be NVE, according to a study on Hg$^0$ oxidation by Br and Cl atoms (Sun et al., 2016). In this study, the Hg(II) in PBM should not be attributed directly to oxidation derived by Br or Cl atoms, because Br and Cl atoms would induce negative odd-MIF in the product Hg(II) during oxidation (Sun et al., 2016) (Fig. 2 process b and c), that was inconsistent with the positive odd-MIF observed in the PBM with Hg(II) as the primary form. When the photo-reduction of aquatic Hg(II)

involving dissolved organic matter occurs, especially in the MBL with high RH values, the direction of



odd-MIF might reverse because positive odd-MIF would be induced in Hg(II) in aquatic layer on particle surfaces (Zheng and Hintelmann, 2009, 2010). The magnitude of photo-reduction should be much greater than oxidation derived by Br/Cl atoms to produce the observed positive odd-MIF in PBM. However, ratios of $\Delta^{199}$Hg/$\Delta^{201}$Hg in PBM measured in this study were much higher than 1.0, that value has been

associated with the photo-reduction of aquatic Hg(II) (Zheng and Hintelmann, 2009, 2010;Bergquist and Blum, 2007). Therefore, Br/Cl atoms-derived photo-oxidation followed by the photo-reduction of aquatic Hg(II) should not be the primary routine leading to the isotopic compositions in PBM in this study. In addition, correlation between Hg isotopic compositions in PBM and speciated Br concentrations on the TSPs was also absent (Table S3). All of these results suggested that Br or Cl atoms were not the direct

contributor to the Hg(II) in PBM in the MBL in this study.

Alternative oxidizers other than Br and Cl atoms, including the derivatives of Br/Cl atoms (e.g., BrO, HOCl, OCl⁻), ozone, hydroxyl radicals, nitrate radicals, and iodine radicals, might play more important roles to the Hg(II) in PBM in this study. The limited isotopic study of Hg⁰ oxidation prevented the specific oxidizers from being identified here. However, the following clues might be helpful to

uncover the oxidizers and oxidation processes in the future. According to the isotopic signatures present in TGM and PBM, the primary oxidation of Hg⁰ by unidentified oxidizer should induce a positive odd-MIF in the Hg(II) (Fig. 2 process d). To date, no evidence suggests that odd-MIF could occur during the adsorption of Hg(II) on the surface of particles (Fig.2 process g), and the high ratios of $\Delta^{199}$Hg/$\Delta^{201}$Hg observed in PBM in this study indicated the insignificance of continental impacts. Therefore, the high

$\Delta^{199}$Hg and high ratios of $\Delta^{199}$Hg/$\Delta^{201}$Hg in the PBM should be primarily attributed to the oxidation of Hg⁰, following the NVE mechanism. The MIF driven by NVE shares the same direction as the MDF induced during certain process (Schauble, 2007). Negative MDF must be subsequently induced in Hg(II) after oxidation, producing the negative $\delta^{202}$Hg and positive $\Delta^{199}$Hg values in the PBM examined here. Lighter isotopes prefer to be bound on the surfaces of particles, that could be the most possible reason to

negative MDF (Fig.2 processes g).

It should be noted that the primary process leading to the Hg(II) in PBM could not be the primary oxidation processes of Hg⁰ in MBL in this study. The GOM attributed to multiple oxidizers would show a variety of occurrence forms, with different adsorption capacities on particle surfaces. The specific forms of Hg(II) on particle surfaces remained unclear in this study. Meanwhile, the oxidation processes leading

to the Hg(II) in PBM might also occur in the interface reaction layer on particle surfaces. To date, the



mechanisms and isotopic fractionations on the formations of GOM were poorly understood, preventing the accurate cognitions to the Hg(II) in PBM.

On the other hand, all slopes obtained by performing a linear regression of the $\Delta^{199}$Hg and $\Delta^{201}$Hg in the TGM samples were higher than 1.0 (Fig. 4). A ~1.0 slope was commonly observed in TGM

collected at sites influenced by anthropogenic emissions (Gratz et al., 2010;Yu et al., 2016;Yin et al., 2013). While lower slopes (0.5–0.8) were observed in TGM collected at remote sites located at island (Fu et al., 2018), coast (Demers et al., 2015;Rolison et al., 2013), forest (Demers et al., 2013;Fu et al., 2018;Yu et al., 2016), and summit (Fu et al., 2016). Among the three cruises, the highest slope was observed in TGM collected in 2016-summer cruise. Meanwhile, the most near-zero $\Delta^{199}$Hg values and

lowest TGM concentrations were also observed during this cruise as mentioned. This result indicated that alternative factors elevated the slope shaped by mixing continental emissions to clean air in MBL. Positive odd-MIF and high slope of $\Delta^{199}$Hg/$\Delta^{201}$Hg (Br: 1.64; Cl: 1.89) could be induced in remained $Hg^0$ pool when oxidation derived by Br/Cl atoms occurred (Sun et al., 2016), consistent to the odd-MIF signatures in TGM in this study. Furthermore, negative MDF would occur in $Hg^0$ pool during the

oxidation derived by Br atoms (Fig. 2 process b), that was also consistent to the negative MDF in TGM collected during two summer cruises, contrasting to the opposite direction of MDF in $Hg^0$ when oxidation was derived by Cl atoms (Fig. 2 process c). Therefore, Br atoms were suggested to be the primary oxidizer for $Hg^0$ in MBL, echoing the demonstration in previous publications (De Simone et al., 2013;Holmes et al., 2010;Holmes et al., 2009;Ye et al., 2016;Obrist et al., 2011).

In addition, potential alternative factors might also contribute to the transformations of TGM in this study, followed by these isotopic clues. A negative correlation (P < 0.01) between $\Delta^{199}$Hg in the TGM and the atmospheric temperature (17.7 to 28.4°C) was observed during the 2018-summer cruise, indicating that process inducing positive odd-MIF in TGM could be more active at lower temperatures, enhancing the oxidation and scavenging of $Hg^0$ in the MBL (Hedgecock and Pirrone, 2004). However,

the correlation was absent during the 2016-summer cruise, which is possibly due to the narrow temperature range involved (22.5 to 24.7°C) (Fig. 3a), and also absent during the 2016-winter cruise with lower temperatures. Despite of the similar large temperature range (-1.4 to 12.0°C), and the similar positive correlation (P = 0.03) between $\Delta^{199}$Hg in TGM and TGM concentration with 2018-summer cruise (Fig. 3b), that correlation absence indicated the process might be weak during winter cruise.





Emissions from surface sea water (Fig. 2 process h) are commonly considered to be crucial to influencing atmospheric Hg in MBL. However, in this study, this factor should be considered less important. Volatilization of dissolved gaseous Hg should induce negative MDF to $Hg^0$ in the MBL (Zheng et al., 2007), which partially contributed to the negative $\delta^{202}Hg$ observed in the TGM. This process should not produce odd-MIF (Zheng et al., 2007). According to the negative correlation observed

between air temperature and $\Delta^{199}Hg$ in TGM in 2018-summer cruise, if the elevated temperature accelerating Hg volatilization from surface sea water was an important factor shaping isotopic compositions in TGM, the similar correlation between $\Delta^{199}Hg$ in TGM and air temperature should also be observed in winter, which was absent in this study.

        Transformations of atmospheric Hg are complicated. The mechanisms and isotopic fractionations

of transformation processes are poorly understood. For instance, the photo-reduction of Hg(II) in gaseous phase (Lin and Pehkonen, 1999;Horowitz et al., 2017) might also induce odd-MIF in the Hg(II) remaining on aerosol surfaces (Fig. 2 process e). On the other hand, some gaseous mercury, e.g., MeHg and diMeHg in plume, have been suggested as important components to atmospheric Hg in MBL (Barkay et al., 2011;Baya et al., 2015), and the Hg isotopic compositions in those components remain unclear

(Fig.2 process k). Effects of these two factors on isotopic compositions in TGM and PBM in the MBL cannot be ruled out.

        Odd-MIF occurrences are commonly associated with photo chemical reactions (Bergquist and Blum, 2007;Sun et al., 2016). However, isotopic compositions in TGM or PBM collected in daytime and nighttime were insignificant different in this study (T-test, P > 0.05). A possible reason is that the isotopic

fractionations caused by photo chemical reactions were diluted due to the low time resolutions during sampling.

**4 Conclusions**

        Our measurements of TGM and PBM samples collected in Chinese MBL suggested Br atoms could be the most possible oxidant to TGM, but alternative oxidants other than Br or Cl atoms play a major

role in the formation of Hg(II) in PBM. These oxidation processes could largely shift the Hg isotopes in the atmosphere, producing negative MDF, positive MIF, and elevated slopes in linear regression results of $\Delta^{199}Hg/\Delta^{201}Hg$ in TGM, as well as more positive MIF and high ratios of $\Delta^{199}Hg/\Delta^{201}Hg$ in PBM



following NVE mechanism. Lower air temperature could promote relevant processes causing positive MIF in TGM in summer, while the relative processes might be weak in winter.

To our knowledge, isotopic fractionation that occurs during Hg environmental processes is diluted by isotopic signatures inherited from multiple emission sources, especially from anthropogenic emissions, and thus has been omitted in previous studies conducted at continental sites when a stable Hg isotopic tracking method was used. In this study, however, the mixing with continental emissions could not entirely lead to the isotopic signatures in atmospheric Hg. The observed isotopic signatures indicated the

importance of local Hg environmental behaviours caused by an abundance of highly reactive species. Therefore, isotopic fractionation occurring during environmental processes should be carefully considered when using stable Hg isotopes to trace sources.

        In this study, isotopic compositions in atmospheric Hg collected from marine areas were different from those collected from most inland areas. Due to the low concentrations of TGM and PBM in the

MBL, the time resolutions of isotopic signatures were low. This would dilute potential isotopic fractionations occurring within each sampling period, e.g., the isotopic fractionation following the GOM concentration increasing associated with air temperature and RH changes, or the potential isotopic diversities associated with the gradient PBM concentration from coastal areas to open seas (Wang et al., 2016a;Wang et al., 2016b). In addition, many atmospheric Hg transformation processes, e.g., the

reduction of Hg(II) in the gaseous phase, are still poorly understood. More studies are therefore needed to constrain isotopic fractionation during these processes. When the sampling and isotopic measurement techniques improve, and the isotopic study of the oxidation of gaseous Hg is performed in the future, stable Hg isotopes could provide diagnostic information for clarifying the contributions of multiple environmental processes influencing atmospheric Hg chemistry, and could serve as effective tools for

tracking transformation processes of atmospheric Hg in the MBL, and in other areas with a variety of atmospheric oxidants in atmosphere.

**Data availability**

Dataset could be accessed in Supplementary Information, or https://doi.org/10.5281/zenodo.3748831.



**Author contribution**

Ben Yu, Lin Yang, Linlin Wang, and Hongwei Liu conducted sampling and measurements; Cailing Xiao

and Yong Liang assisted the measurements; Ben Yu and Jianbo Shi designed research; Ben Yu, Jianbo

Shi, Qian Liu, Yongguang Yin, Ligang Hu, and Guibin Jiang wrote the paper.

**Competing interests**

The authors declare that they have no conflict of interest.

**Acknowledgements**

The authors would thank the captain and the crew of Dongfanghong II for their assistance on sample and

data collection. The authors gratefully acknowledge the NOAA Air Resources Laboratory for the

provision of the HYSPLIT transport and dispersion model and/or READY website

(https://www.ready.noaa.gov) used in this publication. This work was supported by the CAS

Interdisciplinary Innovation Team (JCTD-2018-04), the National Natural Science Foundation of China

(41877367, 91843301, and 21707157), and the Sanming Project of Medicine in Shenzhen

(SZSM201811070).

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



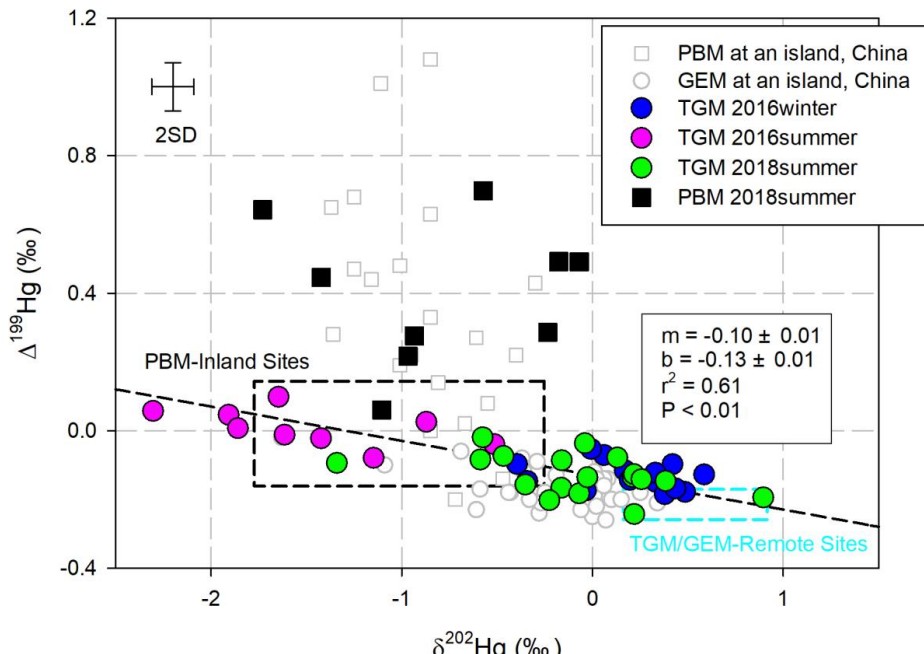

**Figure 1: The scatter plot of δ²⁰²Hg and Δ¹⁹⁹Hg in TGM and PBM samples in this study, illustrated with reported isotopic compositions in GEM (Fu et al., 2018) and PBM (Fu et al., 2019a) collected at Huaniao island site in East China Sea, reported data ranges (mean±SD) of isotopic compositions in TGM/GEM from the remote sites globally (Demers et al., 2015;Yu et al., 2016;Demers et al., 2013;Fu et al., 2016), and in PBM from inland sites (Yu et al., 2016;Das et al., 2016;Huang et al., 2016;Huang et al., 2015;Xu et al., 2017). Error bars refer to max 2σ for samples in this study.**


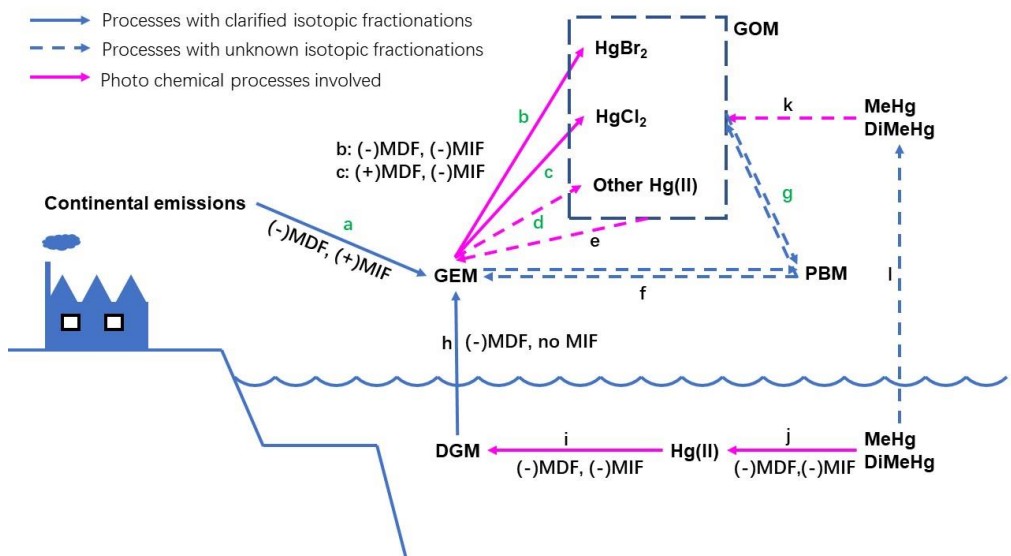

**Figure 2: The atmospheric processes of Hg in MBL with inducing isotopic fractionations (MDF and only odd-MIF directions). a: the mixing with continental emissions; b: oxidation by Br atoms; c: oxidation by Cl atoms; d: oxidations by alternative oxidants; e: photo-reductions of gaseous Hg(II); f: adsorption and desorption of Hg⁰ on airborne particle surfaces; g: adsorption and desorption of Hg(II) on airborne particle surfaces; h: volatilization of dissolved gaseous Hg⁰ from surface sea water; i: photo-reduction of aquatic Hg(II); j: photo-**
**decomposition of aquatic MeHg/diMeHg; k: photo-decomposition of gaseous MeHg/diMeHg; l: volatilization of dissolved MeHg/diMeHg.**

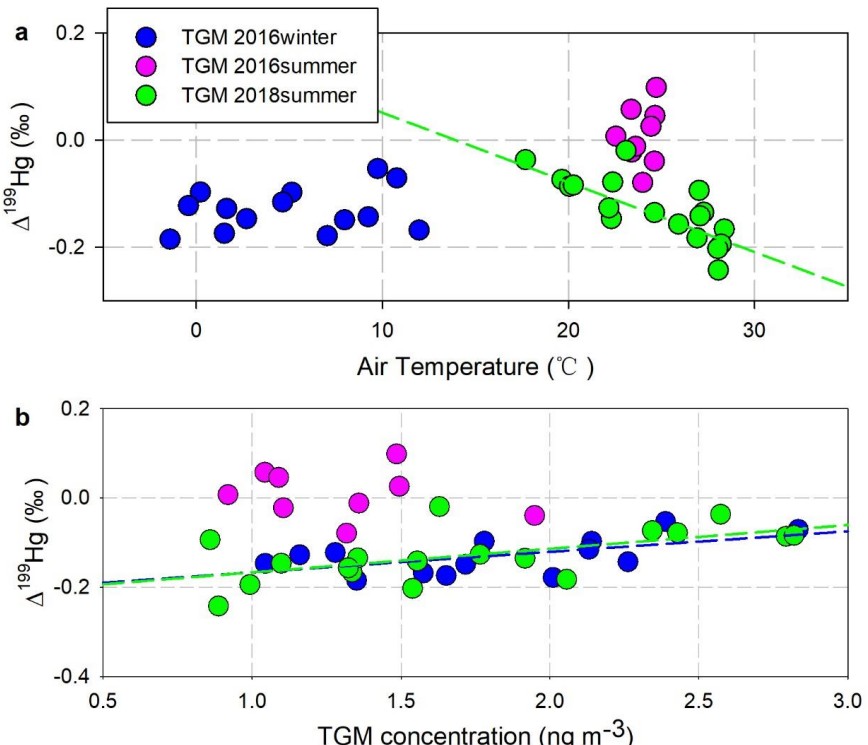

**Figure 3: The correlation between Δ$^{199}$Hg in the TGM and air temperatures (panel a), and between Δ$^{199}$Hg in**
**the TGM and TGM concentrations (panel b). Regressions lines were coloured to match the TGM scatters.**

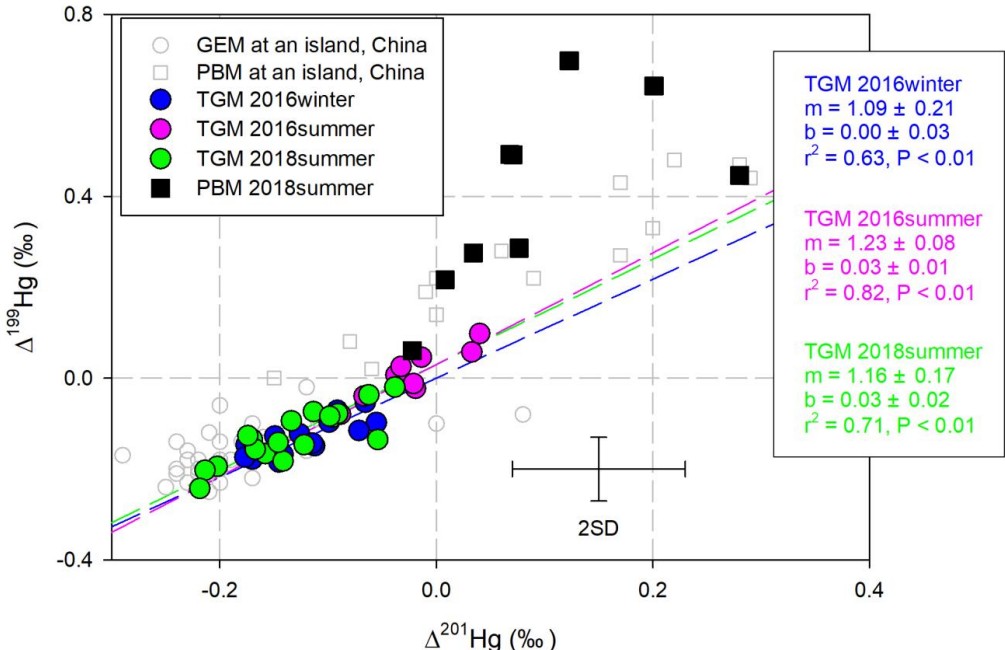

**Figure 4: Scatter plot of $\Delta^{199}$Hg and $\Delta^{201}$Hg in the TGM and PBM samples, illustrated with reported data obtained at Huaniao island sites in East China Sea (Fu et al., 2018;Fu et al., 2019a). Regression lines were coloured to match the TGM scatters. Error bars refer to 2σ for samples in this study.**




Table 1: Statistical summary table of isotopic data in this study. The comparing data from previous publications is characterized to TGM in remote area,(Demers et al., 2013;Demers et al., 2015;Fu et al., 2016;Fu et al., 2018;Yu et al., 2016) GEM in an island located in East China Sea,(Fu et al., 2018) PBM collected at inland sites,(Yu et al., 2016;Das et al., 2016;Huang et al., 2016;Huang et al., 2015;Xu et al., 2017) and PBM collected in a coastal site in United States(Rolison et al., 2013) and the same island in East China Sea(Fu et al., 2019b)

| Sample | Sampling Time | Concentration* ng (pg) m⁻³ | 1σ* ng (pg) m⁻³ | δ²⁰²Hg (‰) | 1σ (‰) | Δ¹⁹⁹Hg (‰) | 1σ (‰) | Δ²⁰⁰Hg (‰) | 1σ (‰) | Δ²⁰¹Hg (‰) | 1σ (‰) |
|---|---|---|---|---|---|---|---|---|---|---|---|
| TGM | 2016-winter | 1.81 | 0.51 | 0.19 | 0.30 | -0.13 | 0.04 | -0.03 | 0.02 | -0.12 | 0.04 |
| TGM | 2016-summer | 1.31 | 0.31 | -1.48 | 0.56 | 0.01 | 0.05 | -0.03 | 0.04 | -0.02 | 0.04 |
| TGM | 2018-summer | 1.74 | 0.64 | -0.09 | 0.48 | -0.13 | 0.06 | -0.05 | 0.04 | -0.13 | 0.05 |
| PBM | 2018-summer | 14.3 | 19.8 | -0.80 | 0.58 | 0.40 | 0.21 | 0.01 | 0.03 | 0.09 | 0.10 |
| TGM/GEM-remote | | | | 0.54 | 0.38 | -0.21 | 0.04 | -0.05 | 0.03 | -0.20 | 0.04 |
| GEM-island | | | | -0.21 | 0.39 | -0.16 | 0.06 | -0.06 | 0.04 | -0.18 | 0.07 |
| PBM-inland | | | | -1.01 | 0.76 | -0.01 | 0.15 | 0.03 | 0.04 | -0.03 | 0.14 |
| PBM-coastal/island | | | | -0.87 | 0.36 | 0.50 | 0.41 | 0.10 | 0.06 | 0.35 | 0.39 |

*: The concentrations reported in this study were calculated by measured THg concentrations in trapping solution and measured air volume during sampling. The TGM concentrations were reported in unit of ng m⁻³, and PBM concentrations were reported in unit of pg m⁻³.