# Peer review of "New evidence for atmospheric mercury transformations in the marine boundary layer from stable mercury isotopes"

_Atmospheric Chemistry and Physics, 2020_

## Referee Comment (RC1) · Anonymous Referee #1 · 27 May 2020

In this study, the authors present a comprehensive study on the isotope compositions of atmospheric total gaseous mercury (TGM) and particulate bound mercury (PBM) in the marine boundary layer over Chinese seas. As I know, this is a first study for investigating the isotope compositions of atmospheric Hg species and their underling mechanisms in MBL. It is therefore novel and would contribute significantly the Hg isotope research field. The manuscript is overall well written and organized. I agree with the interpretations on the mechanisms associated with the variations of Hg isotope compositions. Actually, I have revised this manuscript when it was submitted to another journal. I found the authors have address most of my and other reviewer's comments properly, and the paper is currently in a good quality. I would therefore suggest a publication of the manuscript in ACP with minor revisions. Some of the minor comments

are listed below: 1. line 15: should be 'isotopic compositions of total gaseous mercury (TGM) and particulate bound mercury (PBM) collected. . ..' 2. line 18: should be 'isotopic compositions of TGM in MBL were affected. . .' Line 19-20: the statement of 'lower air temperature could promote the positive. . ." is not clear. I would suggest to specify to 'TGM D199Hg values were significantly positively correlated with air temperature. . .', and then the author may interpret the potential mechanisms. Section 2.1: I would suggest to add a figure to show the cruise and sampling locations of the isotope samples in the main manuscript. Line 150-154: the diagnostics of using D199Hg/d202Hg ratios is great. But I think it should simply introduce the ratios obtained from previous laboratory studies or source materials, and this would help you to figure out the major factors. Line 168: 'TGM collected in two summer cruises . . .ranges' should be " TGM collected in two summer cruises were characterized by significantly negative d202Hg and near-zero D199Hg values. Line 177-179: 'the back-trajectory. . ..in TGM' should reword as 'Backward trajectory analysis showed that higher D199Hg values were associated with air masses originated from both mainland China and open oceans' Line 202-205: need to be rewritten. Note that the MIF signatures of gaseous Hg(II) emitted from anthropogenic sources and produced via atmospheric oxidation should be different. The former case would have near-zero value, while the latter case would have significantly positive value according to precipitation observations. These two should be differentiated here. Line 210-215: the D199Hg/D201Hg ratio for PBM should be presented Line 285-293: I agree with your interpretation that oceanic GEM emission should play a minor role. However, as the isotopic composition of oceanic GEM emissions have not been well constrained, I would suggest to soft these statement. Maybe this process could be completely excluded. Zheng et al., investigated the MDF of Hg isotope during aqueous Hg(0) evaporation. Note that GEM emission from water is driven by evaporation of Hg(0) from water (the isotopic compositions are unknown) and water-atmosphere interface photoreduction (the isotopic compositions of initial water are unknown). So it is difficult to known whether this source generate positive or negative MIF.

---

## Referee Comment (RC2) · Anonymous Referee #2 · 3 Jun 2020

This study reported the isotope compositions of total gaseous mercury (TGM) and particulate bound mercury (PBM) in the marine boundary layer (MBL) during summer and winter seasons. The results are novel and very interesting. It has significant contributions to the research field and improve the understanding of Hg transport and transformation in MBL. The manuscript is also written clearly and well organized. Therefore, I suggest a publication of this manuscript in ACP with minor revisions. My minor comments and questions: 1. Line 202-204, what do the authors mean by "contribution of Hg(II) in wet deposition to both PBM and TGM"? How Hg(II) in wet deposition contribute to PBM and TGM? Do you indicate the sources of Hg(II) in wet deposition were not directly from PBM or TGM? 2. A general comment that conclusions should be made with caution when comparing statistic slopes derived from different studies. As

the air samples are usually mixtures from different sources, and not all sources are well quantified and constrained.

---

## Author Comment (AC1) · 4 Jun 2020

In this study, the authors present a comprehensive study on the isotope compositions of atmospheric total gaseous mercury (TGM) and particulate bound mercury (PBM) in the marine boundary layer over Chinese seas. As I know, this is a first study for investigating the isotope compositions of atmospheric Hg species and their underling mechanisms in MBL. It is therefore novel and would contribute significantly the Hg isotope research field. The manuscript is overall well written and organized. I agree with the interpretations on the mechanisms associated with the variations of Hg isotope compositions. Actually, I have revised this manuscript when it was submitted to another journal. I found the authors have address most of my and other reviewer's comments properly, and the paper is currently in a good quality. I would therefore suggest a

publication of the manuscript in ACP with minor revisions.

Response: We appreciate very much the valuable comments from you and the other previous reviewers. These comments really improved our manuscript. We hope this study could provide useful data and discussion for better understanding the atmospheric Hg transformation and transport, especially in marine boundary layer.

line 15: should be 'isotopic compositions of total gaseous mercury (TGM) and particulate bound mercury (PBM) collected. . .' line 18: should be 'isotopic compositions of TGM in MBL were affected. . .'

Response: Thank you very much for these linguistic modifications. These two modifications have been made in the revised manuscript.

Line 19-20: the statement of 'lower air temperature could promote the positive. . .." is not clear. I would suggest to specify to 'TGM D199Hg values were significantly positively correlated with air temperature. . .', and then the author may interpret the potential mechanisms.

Response: The sentence has been changed to 'D199Hg values of TGM were significantly positively correlated with air temperature in summer, indicating that processes inducing positive odd-MIF in TGM could be more active at low temperatures, while the relative processes might be weak in winter.'

Section 2.1: I would suggest to add a figure to show the cruise and sampling locations of the isotope samples in the main manuscript.

Response: We thank the referee for this suggestion. Indeed, we made three GIF files to show the cruise and wind field during three sampling cruises, respectively. We have uploaded the GIF files to https://doi.org/10.5281/zenodo.3871222 as supporting information. The announcement of these supporting information has been added in the Data Availability part in the revised manuscript.

Line 150-154: the diagnostics of using D199Hg/d202Hg ratios is great. But I think it

should simply introduce the ratios obtained from previous laboratory studies or source materials, and this would help you to figure out the major factors.

Response: We agree with the referee. Description on this ratios and regression slopes reported in references have been added as 'This fitted curve always indicated a mixing of plumes with different isotopic fingerprints (Demers et al., 2015;Yu et al., 2016;Fu et al., 2018). Especially a $\sim$ -0.1 slope could be shaped when mixing of plumes from anthropogenic emissions characterized by negative $\delta$202Hg and near-zero $\Delta$199Hg values, and plumes from remote areas characterized by positive $\delta$202Hg and negative $\Delta$199Hg values, e.g., three slopes of -0.09, -0.13, and -0.07 observed in TGM from Mt. Damei, Mt. Ailao, and Beijing, China, respectively (Yu et al., 2016), and -0.095 observed in TGM/GEM and source materials worldwide (Fu et al., 2018)'

Line 168: 'TGM collected in two summer cruises . . . ranges' should be 'TGM collected in two summer cruises were characterized by significantly negative d202Hg and near-zero D199Hg values. Line 177-179: 'the back-trajectory . . . in TGM' should reword as 'Backward trajectory analysis showed that higher D199Hg values were associated with air masses originated from both mainland China and open oceans'

Response: We thank the reviewer for these comments. The modifications have been made in the revised manuscript.

Line 202-205: need to be rewritten. Note that the MIF signatures of gaseous Hg(II) emitted from anthropogenic sources and produced via atmospheric oxidation should be different. The former case would have near-zero value, while the latter case would have significantly positive value according to precipitation observations. These two should be differentiated here.

Response: Here we use the D200Hg in samples to evaluate the contribution from wet precipitation to TGM and PBM in this study. This method was also used in publications. For instance, Enrico et al. (2016), DOI: 10.1021/acs.est.5b06058, evaluated the contributions from wet precipitation to the Hg deposited in peat bogs. Higher D200Hg

values (∼0.2‰ were observed in wet precipitation worldwide, no matter the study sites were in urban/rural or remote areas, with or without the impact from anthropogenic emissions. Therefore the observed near-zero D200Hg values in TGM and PBM in this study suggested limited contributions from Hg(II) in wet precipitations via photo-reduction and re-emission from droplet surfaces. We have updated these sentences to more accurate description on that.

Line 210-215: the D199Hg/D201Hg ratio for PBM should be presented.

Response: Ratio data (6.8±8.4, 1SD, n = 9) has been added in the revised manuscript.

Line 285-293: I agree with your interpretation that oceanic GEM emission should play a minor role. However, as the isotopic composition of oceanic GEM emissions have not been well constrained, I would suggest to soft these statements. Maybe this process could be completely excluded. Zheng et al., investigated the MDF of Hg isotope during aqueous Hg(0) evaporation. Note that GEM emission from water is driven by evaporation of Hg(0) from water (the isotopic compositions are unknown) and water-atmosphere interface photoreduction (the isotopic compositions of initial water are unknown). So it is difficult to known whether this source generate positive or negative MIF.

Response: We agree with the referee that isotopic compositions in dissolved gaseous Hg in surface sea water are unknown. We have highlighted the unknown fact in the revised manuscript and have deleted the arbitrary discussion in this part.

---

## Author Response (AR1)

**Author's response**

**Response to Referee #1:**

*In this study, the authors present a comprehensive study on the isotope compositions of atmospheric total gaseous mercury (TGM) and particulate bound mercury (PBM) in the marine boundary layer over Chinese seas. As I know, this is a first study for investigating the isotope compositions of atmospheric Hg species and their underling mechanisms in MBL. It is therefore novel and would contribute significantly the Hg isotope research field. The manuscript is overall well written and organized. I agree with the interpretations on the mechanisms associated with the variations of Hg isotope compositions. Actually, I have revised this manuscript when it was submitted to another journal. I found the authors have address most of my and other reviewer's comments properly, and the paper is currently in a good quality. I would therefore suggest a publication of the manuscript in ACP with minor revisions.*

Response: We appreciate very much the valuable comments from you and the other previous reviewers. These comments really improved our manuscript. We hope this study could provide useful data and discussion for better understanding the atmospheric Hg transformation and transport, especially in marine boundary layer.

*line 15: should be 'isotopic compositions of total gaseous mercury (TGM) and particulate bound mercury (PBM) collected…'*
*line 18: should be 'isotopic compositions of TGM in MBL were affected…'*
Response: Thank you very much for these linguistic modifications. These two modifications have been made in the revised manuscript (L16 and L19).

*Line 19-20: the statement of 'lower air temperature could promote the positive…" is not clear. I would suggest to specify to 'TGM D199Hg values were significantly positively correlated with air temperature…', and then the author may interpret the potential mechanisms.*
Response: The sentence has been changed to '$\Delta^{199}$Hg values of TGM were significantly positively correlated with air temperature in summer, indicating that processes inducing positive odd-MIF in TGM could be more active at low temperatures, while the relative processes might be weak in winter' in the revised manuscript (L20-23).

*Section 2.1: I would suggest to add a figure to show the cruise and sampling locations of the isotope samples in the main manuscript.*
Response: Thank you very much for this suggestion. Indeed, we made three GIF files to show the cruise tracks and wind field during three sampling cruises, respectively. We have uploaded the GIF files to https://doi.org/10.5281/zenodo.3871222 as supporting information. The announcement of these supporting information has been added in the Data Availability part in the revised manuscript (L349-350).

*Line 150-154: the diagnostics of using D199Hg/d202Hg ratios is great. But I think it should simply introduce the ratios obtained from previous laboratory studies or source materials, and*

*this would help you to figure out the major factors.*

Response: We agree. Description on this ratios and regression slopes reported in references have been added as 'This fitted curve always indicated a mixing of plumes with different isotopic fingerprints (Demers et al., 2015;Yu et al., 2016;Fu et al., 2018). Especially a ~ -0.1 slope could be shaped when mixing of plumes from anthropogenic emissions characterized by negative $\delta^{202}$Hg and near-zero $\Delta^{199}$Hg values, and plumes from remote areas characterized by positive $\delta^{202}$Hg and negative $\Delta^{199}$Hg values, e.g., three slopes of -0.09, -0.13, and -0.07 observed in TGM from Mt. Damei, Mt. Ailao, and Beijing, China, respectively (Yu et al., 2016), and -0.095 observed in TGM/GEM and source materials worldwide (Fu et al., 2018)' in the revised manuscript (L154-160).

*Line 168: 'TGM collected in two summer cruises ... ranges' should be 'TGM collected in two summer cruises were characterized by significantly negative d202Hg and near-zero D199Hg values.*

*Line 177-179: 'the back-trajectory ... in TGM' should reword as 'Backward trajectory analysis showed that higher D199Hg values were associated with air masses originated from both mainland China and open oceans'*

Response: Thank you for these comments. The modifications have been made in the revised manuscript (L174-175 and L184-186).

*Line 202-205: need to be rewritten. Note that the MIF signatures of gaseous Hg(II) emitted from anthropogenic sources and produced via atmospheric oxidation should be different. The former case would have near-zero value, while the latter case would have significantly positive value according to precipitation observations. These two should be differentiated here.*

Response: Here we use the $\Delta^{200}$Hg in samples to evaluate the contribution from wet precipitation to TGM and PBM in this study. This method was also used in publications. For instance, Enrico et al. (2016, DOI: 10.1021/acs.est.5b06058), evaluated the contributions from wet precipitation to the Hg deposited in peat bogs. Higher $\Delta^{200}$Hg values (~0.2‰) were observed in wet precipitation worldwide, no matter the study sites were in urban/rural or remote areas, with or without the impact from anthropogenic emissions. Therefore the observed near-zero $\Delta^{200}$Hg values in TGM and PBM in this study suggested limited contributions from Hg(II) in wet precipitations via photo-reduction and re-emission from droplet surfaces. We have updated these sentences (L206-212) to more accurate description on that.

*Line 210-215: the D199Hg/D201Hg ratio for PBM should be presented.*

Response: Ratio data (6.8±8.4, 1SD, n = 9) has been added in the revised manuscript (L218-219).

*Line 285-293: I agree with your interpretation that oceanic GEM emission should play a minor role. However, as the isotopic composition of oceanic GEM emissions have not been well constrained, I would suggest to soft these statements. Maybe this process could be completely excluded. Zheng et al., investigated the MDF of Hg isotope during aqueous Hg(0) evaporation. Note that GEM emission from water is driven by evaporation of Hg(0) from water (the isotopic compositions are unknown) and water-atmosphere interface photoreduction (the isotopic*

*compositions of initial water are unknown). So it is difficult to known whether this source generate positive or negative MIF.*

Response: We agree that isotopic compositions in dissolved gaseous Hg in surface sea water are unknown. We have highlighted the unknown fact in the revised manuscript (L300-303) and have deleted the arbitrary conclusion (L294) in this part.

**Response to Referee #2:**

*This study reported the isotope compositions of total gaseous mercury (TGM) and particulate bound mercury (PBM) in the marine boundary layer (MBL) during summer and winter seasons. The results are novel and very interesting. It has significant contributions to the research field and improve the understanding of Hg transport and transformation in MBL. The manuscript is also written clearly and well organized. Therefore, I suggest a publication of this manuscript in ACP with minor revisions.*

Response: Thank you very much for your positive and valuable comments.

*Line 202-204, what do the authors mean by "contribution of Hg(II) in wet deposition to both PBM and TGM"? How Hg(II) in wet deposition contribute to PBM and TGM? Do you indicate the sources of Hg(II) in wet deposition were not directly from PBM or TGM?*

Response: Thank you for pointing this out. Here we try to use $\Delta^{200}Hg$ values to evaluate the contributions of aquatic Hg(II) in atmospheric aquatic phase, e.g., cloud or fog, to both PBM and TGM. Atmospheric aquatic Hg(II) can be re-emitted into gaseous phase after photo-reduction, which has been demonstrated in many publications (e.g., Lyman et al., 2020, https://doi.org/10.1016/j.scitotenv.2019.135575). Since wet precipitation worldwide shares similar higher $\Delta^{200}Hg$ values (~0.2‰), and Hg(II) is the primary form of Hg in wet precipitation, the observed near-zero D200Hg in this study suggested the contribution from atmospheric aquatic Hg(II) following photo-reduction and re-emission to PBM/TGM should be limited. We have updated a more accurate description on that in revised manuscript (L209-212).

*A general comment that conclusions should be made with caution when comparing statistic slopes derived from different studies. As the air samples are usually mixtures from different sources, and not all sources are well quantified and constrained.*

Response: Thank you very much for your kind suggestion. We agree with you that discussion on the mixing of plumes using statistic slopes should be careful. We have added a sentence at the beginning of the discussion on slope 'This fitted curve always indicated a mixing of plumes with different isotopic fingerprints (Demers et al., 2015;Yu et al., 2016;Fu et al., 2018). Especially a ~ -0.1 slope could be shaped when mixing of plumes from anthropogenic emissions characterized by negative $\delta^{202}Hg$ and near-zero $\Delta^{199}Hg$ values, and plumes from remote areas characterized by positive $\delta^{202}Hg$ and negative $\Delta^{199}Hg$ values, e.g., three slopes of -0.09, -0.13, and -0.07 observed in TGM from Mt. Damei, Mt. Ailao, and Beijing, China, respectively (Yu et al., 2016), and -0.095 observed in TGM/GEM and source materials worldwide (Fu et al., 2018)' in the revised manuscript (L154-160). We also highlighted that isotopic fingerprint in all different sources was not well quantified and constrained in the implication part (L340-343).

The revised manuscript has been attached in the next part.

Changes have been marked with red color.

[revised manuscript text omitted]